# Pax4 in Health and Diabetes

**DOI:** 10.3390/ijms24098283

**Published:** 2023-05-05

**Authors:** Jenna Ko, Vivian A. Fonseca, Hongju Wu

**Affiliations:** Section of Endocrinology, Department of Medicine, Tulane University Health Science Center, New Orleans, LA 70112, USA; jko2@tulane.edu (J.K.); vfonseca@tulane.edu (V.A.F.)

**Keywords:** Pax4, islets, β cells, α cells, diabetes

## Abstract

Paired box 4 (Pax4) is a key transcription factor involved in the embryonic development of the pancreatic islets of Langerhans. Consisting of a conserved paired box domain and a homeodomain, this transcription factor plays an essential role in early endocrine progenitor cells, where it is necessary for cell-fate commitment towards the insulin-secreting β cell lineage. Knockout of Pax4 in animal models leads to the absence of β cells, which is accompanied by a significant increase in glucagon-producing α cells, and typically results in lethality within days after birth. Mutations in Pax4 that cause an impaired Pax4 function are associated with diabetes pathogenesis in humans. In adulthood, Pax4 expression is limited to a distinct subset of β cells that possess the ability to proliferate in response to heightened metabolic needs. Upregulation of Pax4 expression is known to promote β cell survival and proliferation. Additionally, ectopic expression of Pax4 in pancreatic islet α cells or δ cells has been found to generate functional β-like cells that can improve blood glucose regulation in experimental diabetes models. Therefore, Pax4 represents a promising therapeutic target for the protection and regeneration of β cells in the treatment of diabetes. The purpose of this review is to provide a thorough and up-to-date overview of the role of Pax4 in pancreatic β cells and its potential as a therapeutic target for diabetes.

## 1. Introduction

Paired box (Pax) genes are a family of genes that encode transcription factors involved in regulating the development and differentiation of various tissues and organs during embryonic development [1,2,3]. Pax genes contain a conserved DNA-binding domain known as the paired box, which was originally identified in Drosophila and characterized by two helix-turn-helix motifs that enable binding to specific DNA sequences [4,5,6]. Many Pax genes also contain a homeobox as a second DNA-binding domain, and an octapeptide motif [1,2,3,5]. Nine Pax genes (Pax1–Pax9) have been identified [6,7]. Among them, Pax4 has the most divergent paired box showing the lowest homology (53–70%) to any of the other paired domains. The most closely related member to Pax4 is Pax6, which shares 70% homology and reaches 77% when including conservative exchanges [6].

Pax4 consists of the paired box domain and the homeodomain without the presence of the octapeptide motif [2,3,8] (Figure 1). It is a transcription factor containing ~350 amino acid residues across species with a predicted molecular weight of ~38 kD. The deduced amino acid sequences exhibit 80% identity between mouse and human Pax4 [9]. Pax4 is predominantly found in the pancreas during early development and it is crucial in directing proper pancreatic islet cell development [10,11]. Pax4 gene mutations are associated with the development of diabetes [12,13,14,15].

## 2. Pax4 Gene Expression in the Pancreas

Pax4 is predominantly expressed in the pancreas during embryonic development with little detected in other tissues except the pineal gland and retina [16,17]. In the pancreas, Pax4 is expressed in endocrine progenitors and later restricted to insulin-producing β cells [9,18,19]. Its dominant expression is only early on in islet morphogenesis with in situ hybridization displaying Pax4 mRNA expression as early as embryonic day 9.5 (E9.5) [18], which declines considerably toward the end of the gestation period [20]. Pax4 protein expression level, however, decreases in adult pancreatic β cells [11,20,21] and is undetectable by either Electrophoretic Mobility Shift Assay (EMSA) or Western blot analysis. Likewise, Pax4 mRNA is not detected by RT-PCR in the mature mouse pancreas, and instead peaks between E13.5 and E15.5 in the fetal mouse pancreas only [11,20]. Pax4 expression is found to be gradually confined to a small subpopulation of expandable β cells within the islets [22].

The regulatory region of the Pax4 gene appears to mediate its specific expression in pancreatic β cells, and it contains a 407 bp region that is highly conserved between human and mouse [23]. It includes an A2 element known from the islet amyloid polypeptide (IAPP) promoter, an inverted E1 element known from the insulin promoter, two identical G rich elements, and other sequence motifs [23,24] (Figure 2). Pax 4 expression is particularly regulated by the A2 and E1 elements in its essential regulatory region that shares high homology between mouse and human [24].

## 3. The Role of Pax4 in the Embryonic Development of Pancreatic Islets: Determination of β Cell Lineage

The islets of Langerhans in the pancreas contain five types of endocrine cells, which include glucagon-producing α cells, insulin-producing β cells, somatostatin-producing δ cells, ghrelin-producing ε cells, and pancreatic polypeptide-producing PP cells [25,26,27,28]. Pax4 is essential for islet cell lineage determination during embryonic development as demonstrated in multiple Pax4 gene knockout (KO) studies. Pax4 deficient mice display stunted growth and dehydration, and they die within 3 days after birth [18]. Histology examination shows these Pax4^−/−^ neonatal mice lack mature β cells and δ cells, but have substantial increases in α cells and ε cells in their pancreas [18,29]. Pax4 appears to act as an early critical switch that directs β cell lineage acquisition but is subsequently downregulated in order to achieve full β cell maturation. Similar phenotypes have also been observed in a rabbit KO model [30]. Inactivation of Pax4 by CRISPR/Cas9 system leads to significantly stunted growth, hyperglycemia, and neonatal death within 4 days of birth. Compared to wildtype and Pax^+/−^ rabbits, the Pax4^−/−^ rabbits have very few insulin-producing β cells, but a large number of glucagon-producing α cells [30]. These results suggest that Pax4 plays a critical role in determining β (and δ) cell lineage.

Interestingly, in the lower vertebrate zebrafish, the orthologous Pax4 gene appears to be dispensable for β and δ cell differentiation [31]. Pax4 is transiently expressed in the pancreas of zebrafish embryos, mostly in endocrine precursor cells. Pax4 knockdown shows no detectable effects on β and δ cell development but results in a significant increase in α cell number. The increase of α cells appears to be attributable to upregulation of the α cell-specific transcription factor Arx as a result of Pax4 knockdown [31]. On the other hand, Arx knockdown significantly reduces α cell number but does not lead to increases in other islet cells including β cells in zebrafish. Mutual repression of Arx and Pax4 is maintained in zebrafish, similarly as in mice [31,32,33]. These data suggest that Pax4 may have acquired its essential function in β and δ cell differentiation during vertebrate evolution, while Arx function has been well preserved throughout the process [31].

## 4. Mechanisms of Pax4 Action in Islet Cell Development

As a transcription factor, Pax4 typically functions as a transcriptional repressor of insulin, glucagon, and expression of other genes by binding to hormone gene promoters via its consensus binding sequence [11,34]. Potential Pax4 binding sites have been identified using EMSA with recombinant Pax4 proteins and various oligonucleotide probes [11]. A bipartite consensus DNA binding sequence for Pax4 is revealed, in which the homeodomain binding site and paired-domain binding site are separated by 15 nucleotides. Pax4 paired-domain recognition sites contain the ANNN(C/T)CACCC consensus sequence, which differs at only one base from the core of Pax6 consensus binding sequences [11]. Pax6 is another Pax family protein that is expressed in pancreatic islet cells. Pax6 is expressed in all mature islet cells, and can act as either a transcriptional repressor or activator. In β cells, Pax6 activates β cell specific genes such as MafA, Nkx6.1, Pdx1 and suppresses other islet cell-specific genes, thus maintaining β cell identity [35]. Inactivation of Pax6 in the pancreas results in reduction of β and α cell numbers, and an increase in ε cells [35,36,37,38]. Both Pax4 and Pax6 can act in competition with each other given the degree of similarity in their binding sequences [11]. In fact, ectopic expression of Pax4 in glucagon-producing cells is found to inhibit Pax6-mediated transcriptional activation of the glucagon gene promoter by competitively binding to G1 and G3 elements, both of which are Pax6 target sites that control glucagon gene transcription [39]. Along with this indirect Pax4-mediated transcriptional inhibition that is facilitated by competing for binding sites with Pax6, there is transcription repression potentially through DNA-binding-independent mechanisms such as protein–protein interaction(s) due to inherent Pax4 repressor function and activity [39]. In addition to repressing glucagon promoter activity in α cell lines, Pax4 in transfected cells can inhibit the insulin promoter [34,40], demonstrating that Pax4 binding site-mediated transcription repression of the insulin and glucagon promoters can be indirect through competition with the transcriptional activator Pax6 or actively independent of Pax6 [11,34]. Moreover, β cells co-transfected with Pax4 and reporter-linked human IAPP gene promoter constructs demonstrate transcriptional repression of the IAPP promoter as well [40].

Furthermore, there is a pivotal interaction between Pax4 and transcription factor Arx. During embryonic development, Arx transcription factor expression is present in precursor α-, β-, and δ cells [33]. While Arx is necessary for α cell development in favor over β- or δ cell development, Pax4 has been found to work antagonistically [31] and is crucial in promoting β- and δ cell fates while simultaneously inhibiting α cell commitment in an opposing manner [33]. As such, α cell differentiation may be the default mechanism for islet cell fate commitment [18].

## 5. The Role of Pax4 in Adult Islet Cells: Proliferation, Survival, and Transdifferentiation

Pax4 is mainly expressed in differentiating β cells during embryonic development and is largely undetectable in adult islets or mature β cells. Nonetheless, Pax4 expression is found in a subpopulation of β cells at birth, which declines with age and correlates with reduced β cell replication [22]. These Pax4^+^ β cells have two defining characteristics that distinguish them from other mature β cells: (1) they promote β cell survival and are more resistant to stress-induced apoptosis; and (2) they have greater proliferative capacity. This Pax4^+^ β cell subpopulation expands during pregnancy, a state with increased β cell replication, suggesting that Pax4 defines an expandable β cell subpopulation within adult islets [22]. The presence of this Pax4^+^ subset also allows for a potential avenue of β cell recapitulation in response to damage and dysfunction, and is thus important in maintaining adult islet plasticity.

Pax4 in adult islets is upregulated under metabolic stress where it promotes β cell survival and proliferation. Obesity and insulin resistance induce compensational β cell expansion. Increase in Pax4 expression is observed in the islets of type 2 diabetic (T2D) human donors, possibly as a compensatory mechanism to improve β cell function [41]. In obese and diabetic mice, the transcription factor SREBP1c plays a key role in regulating β cell compensatory response. SREBP1c activation upregulates its target gene Pax4, which in turn stimulates cell cycle gene expression and β cell proliferation [42]. Other studies have also shown Pax4 promotes β cell survival and proliferation by regulating cell cycle proteins, survival genes and endoplasmic reticulum (ER) integrity in response to stress [42,43,44,45,46]. Pax4 upregulates the anti-apoptotic gene Bcl-xL, thereby protecting against apoptosis in the INS-1E insulinoma cell line [46]. Pax4 also promotes β cell survival via targeted genes that maintain ER homeostasis in an inflammatory environment, as shown in a genetically engineered mouse model in which Pax4 expression is inducible in β cells [44]. In adult mice, induced Pax4 expression in β cells inhibits thapsigargin-induced ER-stress-related apoptosis, reduces insulitis, and confers protection against experimental autoimmune diabetes [44]. The survival effect of Pax4 overexpression appears to be conserved in human islets. Pax4 gene delivery into primary human islets, which is mediated by an adenoviral vector, improves islet cell survival in vitro and in vivo in streptozotocin (STZ)-induced T1D mice following islet transplantation [47].

Using inducible β cell-specific Pax4 transgenic mice, studies have demonstrated that Pax4 overexpression in mature β cells offers protection against stress-induced apoptosis [45]. The Pax4-expressing transgenic mice are more resistant to STZ-induced β cell destruction and hyperglycemia compared to mice overexpressing a diabetes-linked Pax4 mutant variant Pax4R129W [45]. Isolated islets from these mice are protected against cytokine-induced β cell apoptosis with a decrease in cytochrome C release and an increase in Bcl2 expression. In addition, Pax4 overexpression in β cells leads to the repression of mature β cell markers such as MafA, Glut2, and insulin, but does not affect Pdx1 and Pax6 expression [45]. Reduced Glut2 expression explains, at least partially, the protection against STZ-induced β cell destruction in these mice because STZ enters β cells through Glut2. Interestingly, when Pax4 expression persists long-term (≥4 months), it induces β cell de-differentiation and promotes β cell replication, as evidenced by the reduction of mature β cell markers and an increase in BrdU labeling of β cells in the Pax4 transgenic mice [45]. Together, these data confirm that Pax4 in mature β cells promotes cell survival and proliferation.

As Pax4 plays a critical role in specifying β cell lineage during embryonic development, it is of interest to investigate the effects of forced Pax4 expression in other cell types. Using transgenic mice that conditionally and ectopically express Pax4 with different cell-specific promoters, Collombat et al. investigated this matter in detail [48]. They found that forced Pax4 expression in Pdx1^+^ pancreatic progenitor cells or Pax6^+^ endocrine progenitor cells resulted in the formation of oversized islets containing mostly (>90%) β cells at the expense of α, δ, and PP cells, which confirms the role of Pax4 for β cell lineage specification during development. More interestingly, in transgenic mice where Pax4 is ectopically expressed in glucagon-producing α cells, their pancreata also contain oversized islets and β cell hyperplasia. The mice exhibit glucagon deficiency and are resistant to STZ-induced hyperglycemia. Lineage-tracing experiments revealed that a vast majority of newly formed β cells are derived from the original glucagon+ α cells, demonstrating that ectopic Pax4 expression in α cells can force them to adopt β cell lineage [48]. Further studies show that this effect is age-independent, and the glucagon deficiency-induced compensation results in islet hypertrophy [49]. Lineage-tracing studies show that, upon α-to-β cell conversion, HNF1β^+^ pancreatic duct-lining cells undergo epithelial-to-mesenchymal transition (EMT), and re-express the endocrine progenitor cell marker Ngn3^+^. They subsequently adopt an α cell identity (glucagon^+^), but are converted into β-like cells due to Pax4 expression, which results in β cell hyperplasia and islet hypertrophy [49]. Pax4-mediated α-to-β cell transdifferentiation has also been demonstrated using Ad5-mediated transient Pax4 gene overexpression in lineage-tracing experiments, and it appears to occur in primary human islets as well since the number of the transitional insulin^+^glucagon^+^ bi-hormonal cells increases significantly following Pax4 gene delivery [47,50]. Moreover, adult δ cells with ectopic Pax4 expression have also been found to be converted into β-like cells. Transgenic mice containing δ cell-specific Pax4 overexpression have an increase in insulin-positive β cells and an increase in islet number and size, which is accompanied by a downregulation of δ cell marker Hhex [51]. In summary, these studies demonstrate that Pax4 ectopic expression in adult non-β islet cells can transdifferentiate them into insulin-producing β-like cells.

## 6. The Role of Pax4 in the Development of Diabetes

As discussed, Pax4 is essential for the differentiation of β cells during embryonic development and for the survival and expansion of adult β cells under stress. Complete loss of Pax4 function results in the absolute lack of β cells and insulin, causing hyperglycemia at birth, and is lethal as shown in the Pax4 deficient mouse and rabbit models [18,29,30]. In the human population, mutations in the *PAX4* gene are found to be associated with both type 1 and type 2 diabetes mellitus (T1D and T2D) [12,13,52,53], and maturity onset diabetes of the young (MODY) [14,15,54,55,56] (Table 1). The missense mutation of R121W in the paired domain impairs both DNA binding activity and transcription activity [12]. Loss of Pax4 function through this mutation is associated with T2D. Clinical characteristics of patients identified to have this mutation include a family history of diabetes, glucose intolerance, and past or current insulin therapy [12]. Phenotype similarities between carriers of the R121W mutation and ketosis-prone diabetes (KPD) have revealed a *PAX4* gene variation of R133W associated with KPD predisposition in the population of West African descent [13]. In clinical presentation, KPD patients with R133W display low C-peptide levels and drastic changes to their insulin secretory reserve.

Another genetic variant of *PAX4* that is identified in Thai subjects, R164W, is found to be associated with MODY (type 9) [14]. The positively charged arginine residue at position 164 is located within the DNA-binding homeodomain of Pax4. It is evolutionarily conserved across species. The substitution of the charged amino acid by a nonpolar amino acid is a drastic replacement and would be harmful. In vitro studies in islet cell lines demonstrate that the R164W mutation dampens Pax4 transcriptional repressor activity on the insulin and glucagon promoters [14]. It is also found to segregate with diabetes within a family. More recently, a *PAX4* p.Arg164Gln (i.e., R164Q) missense mutation has been detected to segregate with diabetes in a Brazilian family cohort, purportedly the first PAX4-MODY family reported in Brazil [57]. Diabetic patients of the family who carry this mutation are present with heterogeneous clinical manifestations (retinopathy, polyneuropathy, renal disease) and treatment responses [57].

A *PAX4* IVS7-1G>A mutation causes aberrant mRNA splicing and likewise impairs the repressor function of Pax4 on its target gene promoters [15]. In vitro studies demonstrate that β cells over-expressing this Pax4 mutant are susceptible to apoptosis when exposed to stress conditions, such as a high glucose environment. This mutation co-segregates with diabetes in a MODY9 family, in which several members have severe diabetic complications including retinopathy, nephropathy, and end-stage renal failure [15]. Furthermore, a recent case report reveals that an infant patient presenting with polydipsia and polyuria possesses a rare C.487C>T heterozygous missense mutation in the 7th exon region of the *PAX4* gene, leading to an R163W variation that is associated with MODY9 [58].

Additionally, *PAX4* missense variants R192H and R192S have been found to be associated with an increased risk for early-onset T2D in Chinese subjects [59]. Patients who are carriers for R192H and R192S in this study are reported to have decreased C-peptide levels, higher BMI, and higher diastolic blood pressure compared to early-onset diabetic non-carriers. *PAX4* with the R192H polymorphism has diminished repressor function on its target insulin and glucagon promoters when compared to wild-type Pax4, and it is associated with both MODY and early-onset T2D [55]. The *PAX4* missense variant rs2233580 (p.Arg192His, or R192H) has also been identified by an exome-chip association study, which appears to be an Asian-specific variant that is associated with T2D [53].

Unlike in Pax4 deficient animal models, these *PAX4* mutations are not lethal, most likely because they are able to maintain partial Pax4 native functions. Impaired Pax4 functions, however, make these individuals susceptible to diabetes development. Notably, *PAX4* mutations not only render individuals vulnerable to T1D and MODY, but also increase the risk for T2D development, which supports the importance of β cell function in the etiology of all forms of diabetes. Consequently, *PAX4* mutation can be a risk factor contributing to the predisposition of diabetes pathogenesis.

**Table 1 ijms-24-08283-t001:** **PAX4 mutations identified in human populations and their phenotypes.** GSIS: glucose-stimulated insulin secretion. OGTT: oral glucose tolerance test.

*PAX4* Mutation	Associated Types of Diabetes	Phenotypes	References
R121W	T2D	Impaired DNA binding activity; Loss of Pax4 inhibition on Pax6-induced transcription; Family history of diabetes, impaired glucose tolerance, diabetic state varies from mild to severe	[12]
R133W	Ketosis-prone diabetes	Decreased repressor activity; Severe alteration of GSIS; Low C-peptide levels	[13]
R164W	MODY9	Impaired transcriptional repression of insulin and glucagon promoters; Impaired glucose tolerance; Family history of diabetes	[14]
R164Q	MODY9	Heterogeneous clinical manifestation; Age at diagnosis ranges from 24 years to 50 years	[57]
R163W	MODY9	Low-frequency heterozygous missense mutation, likely autosomal dominant; Affects the DNA-binding homeodomain; Infantile-onset; Polydipsia, polyuria, and hyperglycemia	[58]
R192H	Early-onset T2D, MODY9	Impaired repressor activity; Higher hemoglobin A1c levels; Decreased 2 h C-peptide levels in OGTT	[53,55,59]
R192S	Early-onset T2D	Single-nucleotide polymorphism associated with lower fasting C-peptide levels and 2 h C-peptide levels in OGTT; Higher BMI	[59]
P321H	T1D	Reduced repressor activity in α and β cell lines; Impaired DNA-binding activity; Diminished capacity for β cell proliferation; Positive for anti-islet cell antibodies with susceptibility to diabetic state	[52]
IVS7-1G>A	MODY9	Causes aberrant mRNA splicing and Q250 deletion, impairing Pax4 repressor functions and increasing susceptibility to apoptosis at high glucose; Associated with early-onset renal complications	[15]

## 7. Pax4 as a Potential Therapeutic Target for Diabetes Treatment

Current treatments available for diabetes include insulin injection and other antidiabetic medications that can modify insulin secretion, insulin sensitivity or glucose absorption, such as metformin, sulfonylurea, glucagon-like peptide 1 (GLP-1)/GLP-1 receptor-based medicines, and sodium-glucose cotransporter-2 (SGLT2) inhibitors [36,60,61,62,63,64,65,66,67]. However, despite these treatment options, there is no definitive cure for diabetes, especially for T1D. When it comes to glucose regulation and metabolic homeostasis, the administration of exogenous insulin is inferior to physiological insulin both in terms of timing and dosage. Insulin-treated patients are at risk of insulin-associated acute hypoglycemia and other complications. Moreover, lifetime insulin therapy is subject to patient compliance and may significantly affect the patient’s quality of life. Therefore, other viable therapeutic options are still being investigated, particularly those that target β cell neogenesis. Restoring β cell function is essential not only for T1D treatment, but is also beneficial for T2D patients who have developed β cell dysfunction.

As Pax4 is a principal transcription factor for β cell specification during embryonic development, and can promote β cell survival, proliferation, and transdifferentiation from non-β islet cells in adult islets, targeting Pax4 holds promise to restore β cell mass and function [68,69]. Indeed, transgenic mice over-expressing Pax4 in various cells, including islet progenitors, β, α, and δ cells, have shown increased β cell mass and are resistant to STZ-induced T1D [44,45,48,49,51]. However, these transgenic strategies show unwanted adverse effects: glucagon deficiency, β cell hyperplasia, and islet hypertrophy because of continuous β cell formation via differentiation, proliferation, and transdifferentiation, respectively. Therefore, these genetic engineering or transgenic strategies are not suitable for therapeutic purposes, and more manageable approaches targeting Pax4 have been under exploration.

One approach is to use Pax4 to enhance the differentiation of β cells from stem cells. Constitutive expression of Pax4 in mouse embryonic stem cells (ESCs) significantly improves the development of insulin-producing β-like cells, and when transplanted into STZ-induced T1D mice, these cells can normalize their blood glucose [70]. Along the same line, Lin et al. have demonstrated that Pax4 expression in embryoid body-formatted ESCs stimulates pancreatic differentiation-related genes, such as Foxa 2, Mixl 1, Pdx1, Insulin, and somatostatin, and promotes the differentiation of β-like cells [71]. Moreover, Pax4 overexpression in human ESCs (hESCs) has also been shown to enhance the differentiation of β-like cells [72]. Pax4 overexpression in hESCs by stable transfection results in a strong expression of β cell-specific gene transcripts, which include Pdx1, Isl-1, and Insulin, and these cells exhibit C-peptide release in response to insulin secretagogue tolbutamide [72]. Similarly, acute Pax4 gene expression in hESCs mediated by an adenoviral vector is found to reduce glucagon and Arx gene expression, leading to an increase in insulin^+^ (β-like) cells and a decrease in gluagon^+^ (α-like) cells at the end of in vitro differentiation [73]. In addition, Pax4 gene delivery, together with Pdx1, significantly increases the efficiency in differentiating mesenchymal stem cells (MSCs) into β-like cells and forming islet-like clusters [74]. Taken together, these studies demonstrate that Pax4 expression enhances stem cell-based β cell formation, and thus may be incorporated into generating functional β cells in vitro for transplantation into T1D patients.

Another approach is to use Pax4 in adult islets in order to induce β cell regeneration (via transdifferentiation) from other islet cell types, protect existing β cells, and stimulate β cell proliferation [47,50]. It has been shown that Pax4 gene delivery into primary islets of mouse and human origin by an adenoviral vector, Ad5.Pax4, not only induces α-to-β cell conversion, but also promotes β cell survival [47,50]. The Pax4-expressing primary islets display improved glucose-stimulated insulin secretion (GSIS), and when transplanted into STZ-induced T1D mice, enhance the therapeutic outcome of islet transplantation compared to control islets, showing better blood glucose control [47]. Histological evaluation of the Pax4-expressing islet grafts in vivo shows the presence of β-like cells originated from α cells, and TUNEL assay shows significantly less β cell apoptosis than control treatments in the islet grafts [47]. Given de-differentiation associated with prolonged Pax4 expression [45], it may be beneficial to restrict Pax4 activity to a temporary duration with the use of adenoviral vector-based gene delivery, which mimics physiological Pax4 expression in development.

In addition to islet cells, forced Pax4 expression appears to help in reprogramming hepatic cells into β-like cells. For instance, the transcription factor Pdx1 can induce insulin expression in the liver and reduces hyperglycemia in STZ-induced T1D mice when delivered by an adenoviral vector [75]. The efficiency of liver cell-to-β cell reprogramming is significantly improved when co-treated with MafA, NKx6.1 and/or Pax4, all of which are essential transcription factors for β cell development [76,77]. Along the same line, using lentiviral vector-based gene delivery, Tang et al. have shown that Pax4 co-expression with Pdx1 in cultured hepatic cells leads to the activation of various β cell function-related genes, such as Pax6, MafA, Isl-1, and IAPP. These cells, when transplanted into STZ-induced T1D mice, can reverse hyperglycemia, confirming the formation of functional β cells in these hepatic cells [78,79].

## 8. Conclusions and Perspectives

Pax4 is a critical transcription factor that plays a pivotal role in the development of β cells in the pancreas. In addition to its role in development, Pax4 can promote β cell survival, proliferation, and transdifferentiation from other islet cells. Complete inactivation of Pax4 leads to the absolute lack of β cells and severe hyperglycemia, resulting in neonatal death as demonstrated in Pax4 knockout animal models. In humans, Pax4 mutations with functional impairment can predispose individuals to the development of diabetes, serving as a risk factor in diabetes pathogenesis. Therefore, Pax4 holds the potential to be a therapeutic target in developing novel therapies for diabetes treatment.

Current efforts on exploring Pax4 therapeutic utility have been focused on enhancing β cell differentiation from stem cells, regenerating β cells by transdifferentiation or stimulating β cell proliferation and survival, all involving the use of viral vectors for Pax4 gene delivery in vitro. An attractive alternative would be to deliver Pax4 directly into the pancreas so that it can induce β cell regeneration and proliferation in vivo. This will be particularly useful for diabetes patients with Pax4 mutations, as these patients may benefit the most from Pax4 gene therapy. However, it is important to note that currently, there are no clinically available gene delivery methods that can safely and effectively deliver Pax4 into pancreatic islet cells. Future endeavors are needed to resolve this issue.

Although gene therapy for human diabetes has not yet reached the stage for clinical development, as discussed above, there appears to be considerable potential for regeneration of β cells in human T1D. Furthermore, the true prevalence of MODY and other mutations causing T2D is not known, and the condition is often missed in clinical practice, which leads to inappropriate treatment. Since most forms of MODY lead to impaired β cell function, the regeneration of normally functioning β cells could transform patient management.

Another Pax4-targeting approach is to identify small molecules that activate endogenous Pax4 expression in pancreatic islets. Despite the importance of this area of research, there are currently no reports regarding small molecules or compounds that specifically stimulate endogenous Pax4 expression. Any breakthrough in this area of research will have a significant impact in diabetes treatment, particularly for T1D.

Finally, additional research is needed to fully understand the molecular mechanisms underlying Pax4 action. Although it is clear that Pax4 plays a critical role in β cell development and function, its upstream regulators and downstream targets are not completely understood. The genetic and epigenetic regulations, as well as the signaling pathways involved in Pax4 action, also require further investigation.

## Figures and Tables

**Figure 1 ijms-24-08283-f001:**
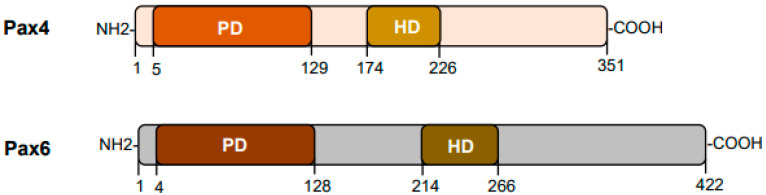
**Diagram of Pax4 protein.** Pax6 structure is included as a comparison. PD: paired domain; HD: homeodomain. The numbers indicate amino acid residue positions in human Pax4 or Pax6 (isoform-a).

**Figure 2 ijms-24-08283-f002:**
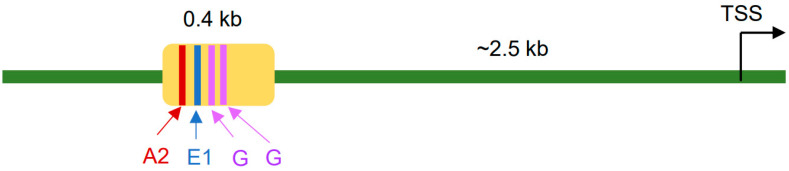
**Diagram of Pax4 promoter and regulatory elements.** A 0.4 kb region (yellow) located at ~2.5 kb upstream of Pax4 transcription start site (TSS) is able to control Pax4 expression in mouse endocrine pancreas, thus representing the minimal (core) Pax4 promoter. The fragment is conserved between mouse and human and contains several regulatory elements including A2, E1, and two G rich elements. A2 and E1 are found in many pancreatic promoters. They can interact with the transcription factors Pdx1 and NeuroD, respectively, and thus are essential in regulating endocrine pancreas gene expression.

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
