# Peer review of "Pax4 in Health and Diabetes"

_ijms, 2023, doi:10.3390/ijms24098283_

Round 1

Reviewer 1 Report

In this review, Ko et al. summarized the expression of Pax4 in the pancreas, the roles of Pax4 in the embryonic and adult pancreatic islets, and the assocation between Pax4 malfunction and the development of Diabetes. The authors believed that Pax4 is a potential therapeutic target for diabetes treatment. It is a well written review, with minor concerns to be addressed.

1. Line 343 " TID" should be "T1D"

2. the readers would appreciate it if the authors could include diagrams showing the roles and underlying mechanisms of Pax4.

Author Response

We appreciate the reviewer's constructive comments. We have revised the manuscript as requested. Specifically: 

1) We corrected the type TID with T1D.

2) We included a Table and a Figure in the revised manuscript. Also included are their legends (Page 23).

Note: Since the Pax4 functions are descriptive, and its molecular mechanisms of action are largely unknown, it is difficult to put them in drawings—it would be either too simplistic, or too complicated with many question markers, thus could be misleading. Nonetheless, we believe the table on Pax4 mutations and the diagram about Pax4 structure (and in comparison with Pax6) are good additions to the manuscript and should be helpful for the readers.

Reviewer 2 Report

In the current review article, Ko et al. highlighted the role of Paired Box 4 (PAX4) in diabetes. The authors mentioned the role of PAX4 in the development of beta cell lineage and islet cell development in the pancreas. The manuscript is very well-written and concise. I still have a few suggestions for the manuscript as mentioned below:

1.     The manuscript lacks the figures. I think the authors should provide a few figures that summarize the text and convey the message clearly. Figures in a review article will make the article more interesting to readers.

2.     Line 342: What is Human TID? I think it is T1D.

3.     Provide the full forms of a few words such as GLP and SGLT2 on line 257.

Author Response

We appreciate the reviewer's constructive comments. We have revised the manuscript as requested. Specifically:

1) We included a Table and a Figure in the revised manuscript. Also included are their legends (Page 23). Since the Pax4 functions are descriptive, and its molecular mechanisms of action are largely unknown, it is difficult to put them in drawings—it would be either too simplistic, or too complicated with many question markers, thus could be misleading. Nonetheless, we believe the table on Pax4 mutations and the diagram about Pax4 structure (and in comparison with Pax6) are good additions to the manuscript and should be helpful for the readers.

2) We corrected the typo TID, and replaced with T1D.

3) The full name for GLP-1 and SGLT2 are included in the revised manuscript.

Reviewer 3 Report

Pax4 gene plays the critical role in beta cells development and function. The genetic and epigenetic regulations and signaling pathways involved in Pax4 action require investigation. The work was prepared methologically correctly. The only remarks are:

1. The purpose of the work is not specified

2. No keywords

As above

Author Response

We appreciate the reviewer's constructive comments, and have revised the manuscript as requested. Specifically:

1) We added a statement about the purpose of the review at the end of the abstract, which is: "The purpose of this review is to provide a thorough and up-to-date overview of the role of Pax4 in pancreatic β cells and its potential as a therapeutic target for diabetes."

2) We included 5 key words on the title page, which are "Pax4, islets, β cells, α cells, diabetes".